# SR$^2$: Boosting 3D Large Language Model with Spatial Relation Reasoning

## Abstract

Recent research in point cloud perception has achieved considerable progress in enhancing scene understanding by means of vision-language alignment through large language models (LLMs). However, existing methods may still encounter challenges in handling complex instructions that require accurate spatial reasoning, even if the 3D point cloud data has provided detailed spatial cues such as size, position, and orientation for identifying the targets. To tackle this issue, this study introduces a new 3D multi-modal LLM framework, Spatial Relation Reasoning (SR$^2$). This framework is designed to strengthen relational reasoning capabilities in 3D environments. SR$^2$ mimics human reasoning behavior by first broadly identifying all relevant elements and then carefully examining them to determine the target. In addition, as current datasets may not comprehensively evaluate the complex spatial reasoning capabilities of various models, we propose a new benchmark named 3D ReasonSeg that consists of 25,000 and 4,152 high-quality samples for training and evaluation respectively. Both quantitative and qualitative experiments demonstrate that SR$^2$ and 3D ReasonSeg effectively endow 3D point cloud perception with stronger spatial reasoning capabilities, and we hope that the proposed SR$^2$ and 3D ReasonSeg can serve as a new baseline and benchmark for future work. The code and model will be made publicly available.

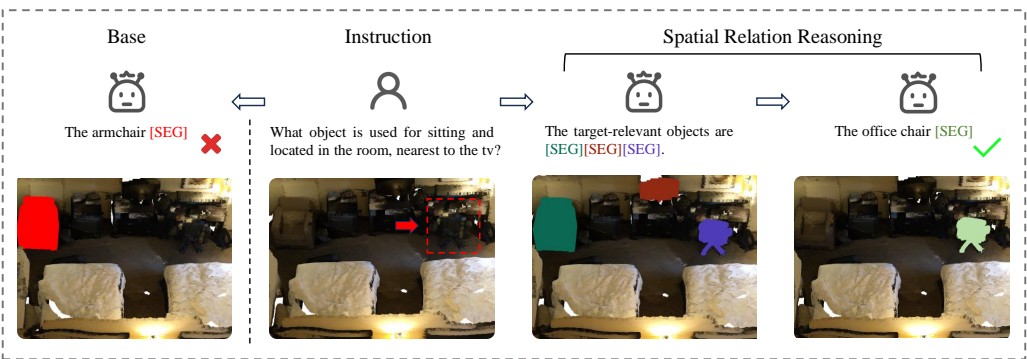

Figure 1: In contrast to the base model in Sec. 3.1, we introduce Spatial Relation Reasoning to explicitly guide the model's reasoning by emphasizing target-relevant objects. In this instance, the object identified for sitting nearest to the TV should be the office chair. The first to fourth columns in the image represent the prediction from the base model, the instruction and ground truth marked by the red rectangle, the predicted target-relevant objects, and the final prediction of SR$^2$, respectively.

## 1 Introduction

In recent years, multi-modal large language models (MLLMs) (Alayrac et al., 2022; Li et al., 2023; Ye et al., 2023; Zhu et al., 2023a; Yang et al., 2023; Wu et al., 2023) have managed to connect language and vision, bringing in a new era of artificial intelligence. These models have shown

unprecedented abilities, enabling various fields to understand and create textual and visual content, such as human-computer interaction, multimedia understanding, embodied intelligence, and more.

Based on the MLLMs, recent works (Chen et al., 2024a; Hong et al., 2023; Wang et al., 2023; Huang et al., 2023a; Xu et al., 2023) have attained outstanding 3D point cloud perception capabilities by offering deeper vision-language reasoning, *i.e.*, 3D MLLMs. However, we observed that the existing 3D MLLM-based methods might not be capable of handling instructions that demand intricate spatial reasoning. As illustrated in Fig. 1, the base model is unable to locate the target even when the relational hints are directly provided. This prompts us to investigate why, compared to the 2D images, the 3D point cloud can provide more accurate and detailed spatial statistics such as size, location, orientation, and relationships with surrounding objects, but the models may still encounter challenges in effectively utilizing this information. We believe that an explicit relational reasoning pipeline is crucial to guide the model in acquiring such reasoning ability.

To this end, we introduce the Spatial Relational Reasoning ($SR^2$) pipeline to enhance the spatial reasoning capabilities of the 3D MLLM through two key steps: 1) Reasoning Prior Learning and 2) Prior-guided Refinement. In the first step, the model is trained to generate spatial priors that offer pertinent clues for the target by identifying the target-relevant objects. Subsequently, in the second step, hidden representations are refined based on these reasoning priors from the first step to aid the spatial reasoning process in pinpointing the final target. In other words, the two steps of $SR^2$ emulate human reasoning behavior, *i.e.*, initially identifying all relevant elements broadly and then scrutinizing them closely to pinpoint the target.

Moreover, we noticed that the queries requiring reasoning in the existing datasets tend to be overly simplistic and ambiguous, as shown in Fig. 5. Consequently, we present a 3D reasoning segmentation dataset called 3D ReasonSeg, which covers object interactions, appearances, and relative spatial relationships. This dataset is designed to improve and comprehensively assess the relational reasoning capabilities of 3D MLLM in 3D scene comprehension. 3D ReasonSeg comprises 25,000/4,152 training/validation samples, respectively, and employs rule-based filtering, including eliminating incorrect spatial relations based on object coordinates, to ensure the data's quality. When trained with the proposed $SR^2$ framework and the 3D ReasonSeg dataset, notable performance enhancements can be observed across popular benchmarks including ScanRefer and ScanQA.

In summary, our contributions are as follows:

- We propose the Spatial Relational Reasoning ($SR^2$) pipeline with two key steps (Reasoning Prior Learning and Prior-guided Refinement) to enhance the spatial reasoning capabilities of 3D MLLM by emulating human reasoning behavior.

- We construct a 3D reasoning segmentation dataset called 3D ReasonSeg to improve and comprehensively assess the intricate relational reasoning capabilities of 3D MLLM in 3D scene comprehension.

- Comprehensive experiments demonstrate the efficacy of the proposed approach. $SR^2$ exhibits the potential to function as a versatile method for enhancing other dense perception frameworks.

## 2 RELATED WORK

### 2.1 3D SCENE UNDERSTANDING

In tasks related to 3D scene understanding, models need to accurately detect scene objects and comprehend their characteristics (Chen et al., 2023; 2022b; Cai et al., 2021; Kolodiazhnyi et al., 2024; Shi et al., 2019; Misra et al., 2021). Among these tasks, 3D segmentation is especially challenging as it involves assigning a label to each point in a point cloud. On the other hand, language can enhance model performance by facilitating their tasks, as demonstrated in 3D Question Answering and 3D Referring tasks (Azuma et al., 2022; Ma et al., 2022; Chen et al., 2020; Zhang et al., 2023; Huang et al., 2022). In these tasks, the model is required to respond to user inquiries or identify objects based on descriptions, highlighting the importance of the model's language comprehension and scenario reasoning abilities.

Also, the integration of language has supported models in transitioning from closed-set to open-set scenarios, such as in 3D Open Vocabulary Segmentation (Peng et al., 2023; Das et al., 2024; Chen et al., 2020; Ding et al., 2023; Takmaz et al., 2023). This step is critical as it allows the model to break free from being constrained by the finite set of categories in the training dataset. While most task-oriented studies focus on specific objectives, recent advancements in 3D multi-modal large language models have enabled addressing multiple 3D downstream tasks using these models (Chen et al., 2024a; Huang et al., 2023b). Furthermore, the assistance of 3D multi-modal large language models has empowered models to tackle more intricate object localization tasks Zhu et al. (2024); Chen et al. (2024b).

However, these studies have not yet addressed the intricate spatial reasoning capabilities needed. In this work, we introduce Spatial Relation Reasoning and 3D ReasonSeg to address complex spatial reasoning challenges.

## 2.2 3D MULTI-MODAL LARGE LANGUAGE MODEL

Significant progress has been made in the realm of 3D multi-modal large language models (3D MLLMs). Initial endeavors predominantly concentrated on point clouds at the object level (Han et al., 2023; Guo et al., 2023; Xu et al., 2023; Liu et al., 2024), posing challenges in tackling complex tasks at the scene level. Recently, there has been a focus on comprehending entire scene point clouds (Chen et al., 2024a; Wang et al., 2023; Huang et al., 2023a; Hong et al., 2023), which has greatly benefited 3D scene understanding tasks.

Inspired by these developments, we also introduce an efficient framework capable of addressing various scene understanding tasks, particularly excelling in 3D scene point cloud segmentation tasks. While these 3D MLLMs can handle fundamental spatial relationships within the scene, they face challenges with intricate spatial connections. In contrast, our model demonstrates proficiency in managing complex spatial relationships.

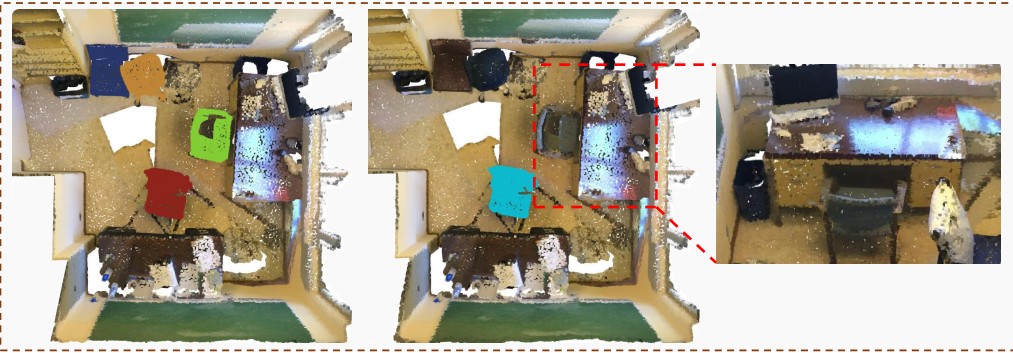

Figure 2: The image on the left depicts the LLM's response to the question "*What is the object that is used for sitting?*" Conversely, the right side of the image illustrates the LLM's response to the question "*What is the object that is used for sitting and located **near the desk with a monitor on it**"* It is noteworthy that the LLM provides an incorrect answer in the latter case. The accurate response should identify the chair encompassed by the red rectangle.

## 3 METHOD

Although efforts such as those in (Chen et al., 2024a; Hong et al., 2023; Huang et al., 2023a) have been made to utilize 3D MLLMs to achieve a better understanding of the 3D point cloud, we observe that the existing methods still have difficulties in handling complex spatial relationships effectively. Specifically, as shown in Fig. 2, 3D MLLM can correctly identify "*what is the object that is used for sitting?*" but face challenges when dealing with queries like "*What is the object that is used for sitting and located near the desk with a monitor on it?*"

To tackle this issue, in this work, we instruct the model to generate reasoning priors by paying attention to those objects relevant to the target. Subsequently, the model examines the scene and

these relevant objects more closely to locate the final target. We refer to this process as Spatial Relation Reasoning.

In the following, Sec. 3.1 will introduce the base model that we have adopted as the baseline. This baseline model can perform dense perception reasonably well when provided with 3D point cloud data. In Sec. 3.2, the details of the proposed Spatial Relation Reasoning will be presented. Subsequently, Sec. 3.3 and Sec. 3.4 will present the specifics of the new benchmark 3D ReasonSeg and the training pipeline, respectively.

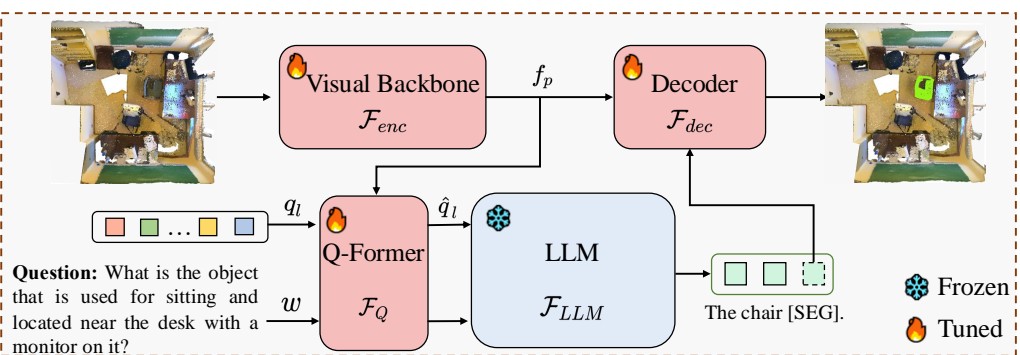

Figure 3: Base model architecture

## 3.1 BASE MODEL

Our base model comprises two parts: a multi-modal large language model (MLLM) and a Segmentation Head. Drawing inspiration from the work of Lai et al. (2024), we have augmented the LLM's vocabulary with a specialized [SEG] token which guides the segmentation head to yield the corresponding segmentation result. This enhancement empowers LLM with sophisticated segmentation capabilities, expanding its functional repertoire beyond language understanding.

**Multi-modal Large Language Model.** There are three key components in the adopted MLLM for feature extraction: a visual encoder, a Q-Former, and a large language model (LLM). Initially, the point cloud $X_p$ is processed through the visual encoder to extract dense features. To optimize computational efficiency, we employ super-point Landrieu & Simonovsky (2018) within the visual backbone, aggregating the dense visual features into super-point features $f_p$ to reduce the following computational burden. Subsequently, the Q-Former Li et al. (2023) compresses the scene's information into several latent queries $q_l$ based on the super-point features $f_p$ and text embedding $w$ of the instruction. After that, the latent queries $\hat{q}_l$ will serve as the visual input, along with the text embedding $w$, to be processed by the large language model $\mathcal{F}_{LLM}$. This process is formulated as:

$$\boldsymbol{f}_p = \mathcal{F}_{enc}(\boldsymbol{X}_p), \quad \hat{\boldsymbol{q}}_l = \mathcal{F}_Q(\boldsymbol{q}_l, \boldsymbol{w}, \boldsymbol{f}_p). \tag{1}$$

$$\boldsymbol{y}_{txt} = \mathcal{F}_{LLM}(\hat{\boldsymbol{q}}_l, \boldsymbol{w}). \tag{2}$$

Inside the Q-Former $\mathcal{F}_Q$, $q_l$ is concatenated with $w$ to be the query in both the self-attention and cross-attention layers of $\mathcal{F}_Q$, while $f_p$ serves as the key and value in the cross-attention:

$$\boldsymbol{q}_l, \boldsymbol{w} = \texttt{SelfAttention}(\texttt{Concat}(\boldsymbol{q}_l, \boldsymbol{w})), \tag{3}$$

$$\boldsymbol{q}_l, \boldsymbol{w} = \texttt{CrossAttention}(\texttt{Concat}(\boldsymbol{q}_l, \boldsymbol{w}), \boldsymbol{f}_p, \boldsymbol{f}_p). \tag{4}$$

**Segmentation Head.** If the output of LLM, *i.e.*, $\boldsymbol{y}_{txt}$, contains [SEG] tokens, it means that the current query requires segmentation predictions. Then, the corresponding hidden features of the [SEG] token, *i.e.*, $\boldsymbol{h}_{seg}$, are processed by the segmentation head $\mathcal{F}_{dec}$ that follows the decoder structure of Kirillov et al. (2023) to yield the requisite segmentation mask $M$, based on the reasoning between $\boldsymbol{h}_{seg}$ and $\boldsymbol{f}_p$. The process can be formulated as:

$$\boldsymbol{M} = \mathcal{F}_{dec}(\boldsymbol{h}_{seg}, \boldsymbol{f}_p). \tag{5}$$

The above framework effectively integrates dense visual perception capabilities into the language model's output, allowing for the fusion of linguistic and visual information processing.

## 3.2 SPATIAL RELATION REASONING

The base model can decently handle dense point cloud perception based on user queries. However, in our observations, the direct instruction tuning performed in Lai et al. (2024) did not consider the spatial relationships among objects in the scene, which hinders the ability to engage in complex spatial reasoning among these objects, as illustrated in Fig. 9. Hence, to address this issue, we suggest that the model initially identifies the objects related to the target, serving as the reasoning prior to guide the target localization. This process is thus termed as Spatial Relation Reasoning (SPR), and it contains two steps as follows.

**Step 1: Reasoning Prior Learning**. While the model may encounter challenges in understanding intricate inter-object relationships, it can recognize objects relevant to the target. As depicted in Fig. 2, despite the model's limitation in accurately segmenting the chair beside the desk supporting a monitor, it demonstrates the ability to detect desks and chairs close to desks. Given this capability, we may instruct the model to initially segment approximate question-related objects, thus establishing the visual reasoning prior as the hint to help identify the final target.

In particular, following the process of the base model, we may first utilize an instruction structured as: "Given the 3D scene, the question is [QUESTION]. Please segment the question-related objects" [1]. The corresponding instruction feature $w$ will be processed by Eq. 1 and Eq. 2 to obtain the text output $y_{txt}$ from which we can obtain the [SEG] tokens representing these target-relevant objects. Subsequently, the segmentation head decodes the hidden features $h_{seg}$ of [SEG] tokens into masks $M_r$ for these target-relevant objects.

After that, we adopt mask pooling to aggregate the super-point features $f_p$ with masks $M_r$, yielding the representations $f_r$ for the target-relevant objects. Both the masks $M_r$ and features $f_r$ of the target-relevant objects will serve as the reasoning priors in the subsequent step. The above process can be formally written as:

$$M_r = \mathcal{F}_{dec}(f_p, h_{seg}), \quad f_r = \texttt{MaskPooling}(f_p, M_r) \tag{6}$$

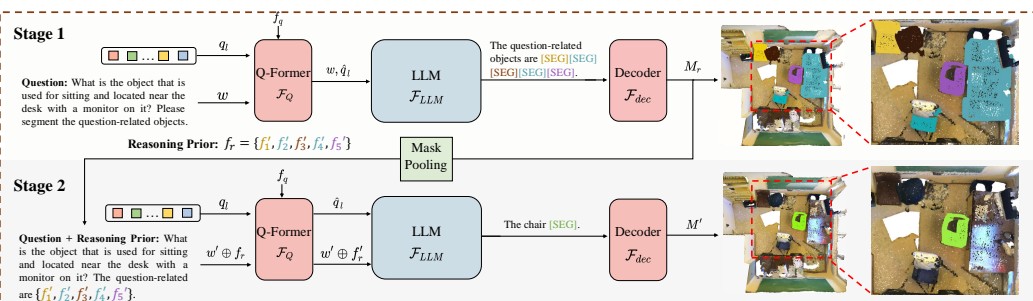

Figure 4: The picture of Spatial Relation Reasoning.

**Step 2: Prior-guided Refinement.** In the absence of explicit instructions, the latent queries $\hat{q}_l$ derived from Eq. 1 may exhibit a bias towards irrelevant areas, potentially impacting the reasoning process in Eq. 2. Besides, the mask-pooling operation in Eq. 6 may cause the loss of essential spatial details. Therefore, we propose the prior-guided refinement which commences by feeding the initial latent queries $q_l$ and target-relevant features $f_r$ into the Q-Former $\mathcal{F}_Q$ to yield the refined query $q'_l$ and target-relevant features $f'_r$.

Specifically, to make use of the obtained reasoning priors, we employ a revised instruction: "Given the 3D scene, the question is [QUESTION]. Please answer the question and output segmentation mask. The question-related

---

[1]More templates are shown in the supplementary file

`objects are {`$\boldsymbol{f}_r^1$`,` $\boldsymbol{f}_r^2$`, ...,` $\boldsymbol{f}_r^n$`}, and you may need to pay attention to them`" [2] where the target-relevant features $\boldsymbol{f}_r$ of $n$ potentially relevant objects have been inserted to form the new instruction whose text embedding is therefore denoted as $\boldsymbol{w}' \oplus \boldsymbol{f}_r$. To this end, the prior-guided refinement regarding the latent query and the target-relevant features can be formulated as:

$$\boldsymbol{q}_l', \boldsymbol{f}_r' = \mathcal{F}_Q(\boldsymbol{q}_l, \boldsymbol{f}_p, \boldsymbol{w}' \oplus \boldsymbol{f}_r) \tag{7}$$

where $\boldsymbol{w}'$ and $\boldsymbol{f}_p$ are the text embeddings and super-point features, respectively. Different from Eq. 3, during the prior-guided refinement, $\boldsymbol{q}_l$ is concatenated with the instruction embedding $\boldsymbol{w}' \oplus \boldsymbol{f}_r$ to serve as the queries for both self-attention and cross-attention layers in the Q-Former $\mathcal{F}_Q$. Meanwhile, $\boldsymbol{f}_p$ continues to function as the key and value in the cross-attention layer:

$$\boldsymbol{q}_l, \boldsymbol{w}' \oplus \boldsymbol{f}_r = \texttt{SelfAttention}(\texttt{Concat}(\boldsymbol{q}_l, \boldsymbol{w}' \oplus \boldsymbol{f}_r)) \tag{8}$$

$$\boldsymbol{q}_l, \boldsymbol{w}' \oplus \boldsymbol{f}_r = \texttt{CrossAttention}(\texttt{Concat}(\boldsymbol{q}_l, , \boldsymbol{w}' \oplus \boldsymbol{f}_r), \boldsymbol{f}_p, \boldsymbol{f}_p) \tag{9}$$

So far, in Eq. 7, the enhanced latent query $\boldsymbol{q}_l'$ has been enriched with target-related clues from $\boldsymbol{f}_r$. Simultaneously, utilizing the point-cloud features $\boldsymbol{f}_p$, the target-relevant features $\boldsymbol{f}_r$ are refined with essential spatial intricacies to produce $\boldsymbol{f}_r'$. Subsequently, the refined query $\boldsymbol{q}_l'$ and target-relevant features $\boldsymbol{f}_r'$ are adopted for generating the new text output $\boldsymbol{y}_{txt}'$ with the LLM $\mathcal{F}_{LLM}$:

$$\boldsymbol{y}_{txt}' = \mathcal{F}_{LLM}(\boldsymbol{q}_l', \boldsymbol{w}' \oplus \boldsymbol{f}_r'), \tag{10}$$

where the instruction embedding $\boldsymbol{w}' \oplus \boldsymbol{f}_r$ originally used in Eq. 7 has been accordingly updated to $\boldsymbol{w}' \oplus \boldsymbol{f}_r'$. Then, the hidden features $\boldsymbol{h}_{seg}'$ of the [SEG] tokens can be extracted from the new text output $\boldsymbol{y}_{txt}'$, followed by the decoder $\mathcal{F}_{dec}$ to generate the refined final mask prediction $\boldsymbol{M}'$:

$$\boldsymbol{M}' = \mathcal{F}_{dec}(\boldsymbol{h}_{seg}', \boldsymbol{f}_p) \tag{11}$$

**Training prior sampling.** The proposed SPR utilizes the mask predictions of relevant objects as the reasoning prior to steer the enhancement of the latent query and text embedding. Nonetheless, we have been observed that directly employing this approach could result in overfitting during model training. This arises because the relevant objects from the text ground truth remain constant during training, while in inference, the model struggles to identify all relevant objects, leading to a disparity between training and testing scenarios.

To alleviate this issue, we introduce a strategy during training where we randomly omit between zero and half of the associated objects. This approach aims to inject diversity into the training dataset, simulating scenarios where the model may encounter incomplete reasoning priors during inference.

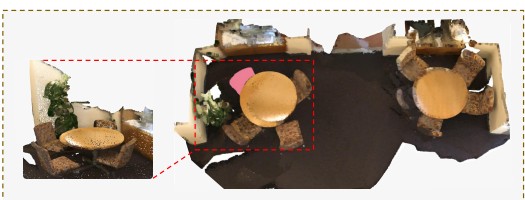

Figure 5: This exemplifies a data sample from ScanRefer, wherein the target object's description reads: "This is a brown chair. It is situated in front of a plant" The spatial relationship delineated in this description exhibits ambiguity, as multiple chairs are positioned anterior to the plant.

### 3.3 DATA CONSTRUCTION

As illustrated in Fig. 5, existing benchmarks include spatial relations like "in front of," which can be simplistic and possibly unclear. To thoroughly assess the ability of 3D large language models to understand detailed and complex spatial relationships, in this section, we introduce a benchmark called 3D ReasonSeg, tailored for detailed spatial reasoning tasks within the 3D point cloud. Later experiments in Sec. 4.3 demonstrate that training with the provided 3D ReasonSeg training set significantly enhances both the baseline model and the model incorporating the proposed SR[2].

---

[2] More templates are shown in the supplementary file.

**3D ReasonSeg.** Specifically, the proposed 3D ReasonSeg requires the model to identify the target object based on a question that does not explicitly mention it. The model needs to understand the visual characteristics, functions, and relationships of the objects described in the query and then use reasoning to segment the target object. To implement this task, we utilize a template structured as: "**USER:** `Given the 3D scene, please provide a segmentation mask in response to the question: [QUESTION]`. **ASSISTANT:** `[ANSWER]`".

We construct the 3D ReasonSeg dataset by combining the detailed scene understanding dataset SceneVerse Jia et al. (2024) with LLama 3.1 Dubey et al. (2024). LLama 3.1 is used to generate question-reasoning-answer sets based on 3D spatial information of objects, such as their positions and dimensions obtained from point cloud data, along with appearance and relational attributes from SceneVerse.

After collecting a considerable number of question-reasoning-answer pairs, we apply rule-based filtering (*e.g.*, calculate the distance between objects and remove the outliers accordingly.) to remove the instances where the targets have unclear and incorrect spatial relationships. To this end, the proposed 3D ReasonSeg dataset comprises 29,152 high-quality data samples. An example is provided in Fig. 6, and additional implementation details, such as the utilized prompts and rule-based filtering, can be found in the supplementary material.

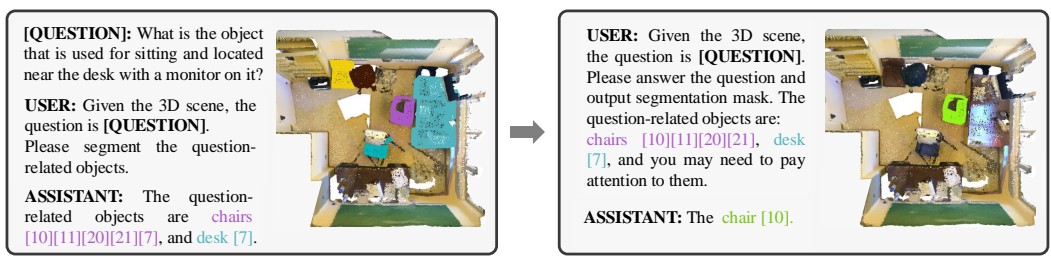

Figure 6: A representative data sample from 3D ReasonSeg. The numerical annotations in the image correspond to unique instance identifiers for each object.

---

**3D Reasoning Segmentation:**
- **USER:** Given the 3D scene, please answer the question with segmentation mask: [QUESTION]. **ASSISTANT:** [ANSWER]
- **USER:** Given the 3D scene, answer this question and output segmentation mask: [QUESTION]. **ASSISTANT:** [ANSWER]

**3D Question Answering:**
- **USER:** Given the 3D scene, please answer the question: [QUESTION]. **ASSISTANT:** [ANSWER]
- **USER:** Answer the question based on the 3D scene: [QUESTION]. **ASSISTANT:** [ANSWER]

**3D Instance Segmentation:**
- **USER:** Given the 3D scene, please segment the [CATEGORY]. **ASSISTANT:** [ANSWER]
- **USER:** Please segment the [CATEGORY] in this 3D scene. **ASSISTANT:** [ANSWER]

**Scene Description:**
- **USER:** Describe the room. **ASSISTANT:** [ANSWER]

**3D Referring Segmentation:**
- **USER:** Given the 3D scene, please segment this object: [DESCRIPTION]. **ASSISTANT:** [ANSWER]
- **USER:** Following are descriptions of a object: [DESCRIPTION]. Please segment it. **ASSISTANT:** [ANSWER]

---

Figure 7: Data formulation for general pre-training.

## 3.4 MODEL TRAINING

**Two-phase training pipeline.** As there is no widely adopted pre-trained vision-language foundation model for point cloud perception, similar to Chen et al. (2024a), we commence by training our model on a broad range of general datasets, including ScanQA Azuma et al. (2022), ScanRefer Chen et al. (2020), ScanNet200 Dai et al. (2017), the ScanNet subset of 3D-LLM Hong et al. (2023), and

the proposed 3D ReasonSeg. As shown in Fig. 7, for different tasks, we adopt different templates where the instructions and answers are given by the USER and ASSISTANT, respectively.

Then, we instruct the model to acquire the ability to conduct spatial relation reasoning as outlined in Sec. 3.2. In contrast to conventional point cloud datasets that solely necessitate [QUESTION] and [ANSWER], the proposed approach requires identifying target-relevant objects, as the reasoning prior to guide the spatial relation reasoning. Specifically, by utilizing the annotations of the objects present in the scene and the question-answer pairs from the original dataset, we utilize LLama 3.1 Dubey et al. (2024) to identify the target-relevant objects. The example of these target-relevant data is shown in Fig. 8, and the utilized prompt and more details are shown in the supplementary file.

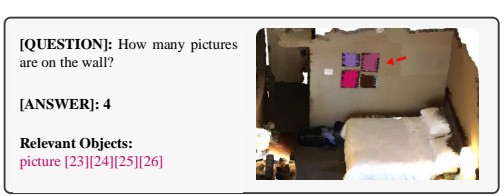 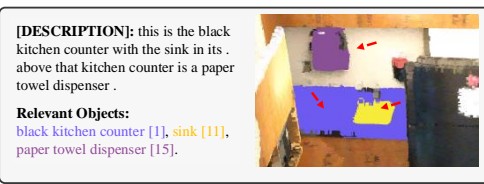

Figure 8: Examples of target-relevant data. The first row depicts an example from ScanQA, while the second row pertains to ScanRefer. On the right of the picture are visualization of target-relevant objects.

In both phases, the model is optimized using the auto-regressive cross-entropy loss $\mathcal{L}_{txt}$ for text generation, alongside the binary cross-entropy (BCE) and DICE losses ($\mathcal{L}_{bec}$ and $\mathcal{L}_{dice}$) for segmentation. Hence, the overall training objective $\mathcal{L}$ can be expressed as:

$$\mathcal{L} = \mathcal{L}_{txt} + \mathcal{L}_{bec} + \mathcal{L}_{dice} \tag{12}$$

## 4 EXPERIMENTS

### 4.1 EXPERIMENTAL SETTING

**Implementation Details.** We adopt the encoder of a pre-trained Oneformer3D as visual backbone and the decoder of it as the mask decoder, and a pre-trained OPT-1.3B Zhang et al. (2022) as the language model. To preserve the learned knowledge of the pre-trained language model, we completely freeze the parameter of the language model while fully training the parameters of all other modules. Note that, we found it is necessary to also fine-tune the visual backbone. We adopt the AdamW Loshchilov (2017) optimizer with a weight decay of 0.1 and a learning rate decaying from 1e-4 to 1e-6 with a cosine annealing scheduler. We randomly sample 300k points from each 3D scene as the 3D input. We conduct experiments on eight Nvidia A100 (80G) GPUs.

**Datasets and Evaluation Metrics.** In this study, we assess our methodologies on ScanQA Azuma et al. (2022), ScanRefer Chen et al. (2020), and our novel 3D ReasonSeg dataset. ScanQA is a dataset for the 3D question answering task, evaluating visual and spatial comprehension of 3D environments. The evaluation metrics of ScanQA include BLEU-4, CIDEr, METEOR, and Rouge-L. ScanRefer evaluates the capability to localize 3D objects using natural language descriptionsthe. Prediction accuracy is measured by the percentage of predictions with Intersection over Union (IoU) above 0.25 and 0.5 compared to ground truth masks. For 3D ReasonSeg, the model is required to focus on object attributes and relationships to reason, answering questions and localizing target objects. Following Lai et al. (2024), the general Intersection over Union (gIoU) metric is used, defined as the average of all per-scene Intersection-over-Unions (IoUs). Detailed formulations of these tasks can be found in the supplementary material.

### 4.2 MAIN RESULTS

To demonstrate the effectiveness of our model, we conduct comparisons with state-of-the-art methods, including specialist models tailored for specific tasks, as well as generalist models capable of addressing multiple tasks.

As indicated in Table 1, our base model has surpassed the majority of existing approaches on both ScanRefer and ScanQA. Notably, in terms of Acc@25 on ScanRefer, our model exhibits significant

Table 1: Results among our method and previous related works. Entries in gray denote models specialized for specific datasets.

| Method | ScanRefer | | ScanQA | | | | 3D ReasonSeg |
|---|---|---|---|---|---|---|---|
| | Acc@25 | Acc@50 | B-4 ↑ | C ↑ | M ↑ | R ↑ | gIoU |
| ScanRefer Chen et al. (2020) | 37.3 | 24.3 | – | – | – | – | – |
| MVT Huang et al. (2022) | 40.8 | 33.3 | – | – | – | – | – |
| 3DVG-Trans Zhao et al. (2021) | 45.9 | 34.5 | – | – | – | – | – |
| ViL3DRel Chen et al. (2022a) | 47.9 | 37.7 | – | – | – | – | – |
| M3DRef-CLIP Zhang et al. (2023) | 51.9 | 44.7 | – | – | – | – | – |
| ScanQA Azuma et al. (2022) | – | – | 10.1 | 64.9 | 13.1 | 33.3 | – |
| 3D-VisTA Zhu et al. (2023b) | 50.6 | 45.8 | 13.1 | 72.9 | – | – | – |
| LLM-Grounder Yang et al. (2024) | 17.1 | 5.3 | – | – | – | – | – |
| 3D-LLM(Flamingo) Hong et al. (2023) | 21.2 | – | 7.2 | 59.2 | 12.2 | 32.3 | – |
| 3D-LLM(BLIP2-flant5) Hong et al. (2023) | 30.3 | – | 12.0 | 69.4 | 14.50 | 35.7 | – |
| Chat-3D Wang et al. (2023) | – | – | 6.4 | 53.2 | 11.9 | 28.5 | – |
| Chat-3D v2 Huang et al. (2023a) | 35.9 | 30.4 | 7.3 | 77.1 | 16.1 | **40.1** | – |
| LL3DA Chen et al. (2024a) | – | – | 13.5 | 76.8 | 15.9 | 37.3 | – |
| Grounded 3D-LLM Chen et al. (2024b) | 47.9 | 44.1 | 13.4 | 72.7 | – | – | – |
| ScanReason Zhu et al. (2024) | 53.1 | 41.1 | – | – | – | – | – |
| baseline | 58.8 | 44.2 | 12.4 | 74.9 | 15.3 | 36.8 | 32.1 |
| + SR$^2$ | **60.0** | **44.3** | **13.9** | **80.3** | **16.4** | 38.4 | **33.6** |

superiority over both specialized and generalized models. Despite maintaining an edge over state-of-the-art generalized models, we acknowledge the potential for enhancement in the Acc@50 metric, suggesting a trade-off between recall and precision due to the constraints of the pre-trained decoder. For ScanQA, our base model demonstrates comparable performance to the leading model Chat3D v2 that lacks segmentation capabilities. Additionally, our base model achieves commendable results on the proposed 3D ReasonSeg task. The experiments demonstrate that our base model serves as a robust 3D MLLM.

While our base model is already highly potent, the incorporation of our proposed SR$^2$ has led to notable enhancements across all metrics, showcasing the efficacy and versatility of our proposed approach. Significantly, the base model enhanced by SR$^2$ has surpassed the top model on ScanQA, a feat that the base model alone could not achieve.

## 4.3 ABLATION STUDY

Table 2: Ablation study on the efficacy of SR$^2$ and 3D ReasonSeg. In this context, 3D RS represents the 3D ReasonSeg.

| Exp. ID | 3D RS | SR$^2$ | ScanRefer | | ScanQA | | | | 3D ReasonSeg |
|---|---|---|---|---|---|---|---|---|---|
| | | | Acc@25 | Acc@50 | B-4 ↑ | C ↑ | M ↑ | R ↑ | gIoU |
| 1 | | | 47.1 | 33.1 | 12.9 | 74.2 | 15.3 | 36.7 | 26.3 |
| 2 | ✓ | | 58.8 | 44.2 | 12.4 | 74.8 | 15.3 | 36.9 | 32.1 |
| 3 | | ✓ | 51.3 | 37.0 | 13.1 | 74.9 | 15.6 | 36.5 | 27.4 |
| 4 | ✓ | ✓ | **60.0** | **44.3** | **13.9** | **80.3** | **16.4** | **38.4** | **33.6** |

**Effectiveness of SR$^2$ and 3D ReasonSeg.** To evaluate the effectiveness of our proposed SR$^2$ and 3D ReasonSeg, we conducted an ablation analysis on ScanRefer, ScanQA, and 3D ReasonSeg. Here, 3D RS refers to the use of 3D ReasonSeg in pre-training, while SR$^2$ indicates the incorporation of SR$^2$ in fine-tuning. The results demonstrate that both SR$^2$ and 3D ReasonSeg individually provide advantages across all datasets, particularly in the case of ScanRefer. Furthermore, the experiments reveal that the combination of 3D ReasonSeg and SR$^2$ synergistically enhance performance, yielding additional benefits.

**Number of latent queries.** We also examined the influence of the quantity of Q-Former latent queries. We experimented with 16, 32, and 64 latent queries, and observed that using 32 latent queries in the default setup leads to satisfactory results.

Table 3: Ablation study on the number of latent queries. The results are predicated on our base method.

| number of latent queries | ScanRefer | | ScanQA | | | | 3D ReasonSeg |
|---|---|---|---|---|---|---|---|
| | Acc@25 | Acc@50 | B-4 ↑ | C ↑ | M ↑ | R ↑ | gIoU |
| 16 | 57.6 | 42.3 | 11.9 | 73.6 | 14.8 | 35.7 | 31.5 |
| 32 | 58.8 | **44.2** | **12.4** | **74.9** | **15.3** | 36.8 | 32.1 |
| 64 | **59.2** | 43.9 | 12.1 | 74.5 | 15.2 | **37.0** | **32.3** |

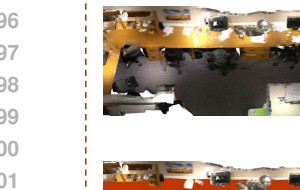
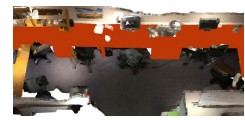
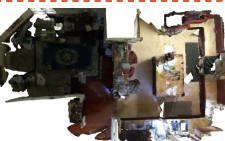
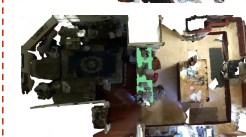

Figure 9: Visual qualitative results on ScanQA and 3D ReasonSeg. Note that the $SR^2$ in the picture is our proposed Spatial Relation Reasoning.

## 4.4 QUALITATIVE RESULTS

As shown in Fig 9, a comparison between the baseline and the baseline with $SR^2$ is presented. On the left side of the image, two qualitative outcomes for ScanQA are displayed. Each red rectangle consists of two rows: the first row shows the prediction from the base model, while the second row displays the prediction from the base model with $SR^2$. The segmentation results in the second row correspond to target-relevant objects. On the right side of the image, one qualitative result for 3D ReasonSeg is presented in a similar format, where the first row represents the base model prediction and the second row illustrates the prediction from the base model with $SR^2$. The bottom left of the blue rectangle shows the segmentation result of the target-relevant objects. More qualitative results are given in the supplementary material.

## 5 CONCLUSIONS

In this work, we have introduced a high-quality dataset, namely 3D ReasonSeg, tailored for the 3D reasoning segmentation task, aiming to enhance and thoroughly evaluate the complex relational reasoning abilities of 3D MLLM in 3D scene comprehension. Additionally, we have proposed an effective framework designed to tackle a range of 3D scene understanding tasks including 3D segmentation. Lastly, we have outlined the Spatial Relation Reasoning pipeline, which serves to guide the model's reasoning explicitly by focusing on target-relevant objects. We hope our work can shed new light on the direction of exploring the spatial reasoning ability of 3D MLLM in the future.

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

## A   DATA CONSTRUCTION DETAILS.

**Data construction prompts.** The prompt used for data generation is illustrated in Fig. 10. We instruct LLama3.1 Dubey et al. (2024) to create question-reasoning-answer pairs and specify the target-relevant objects in the intermediate reasoning steps. The prompt used to construct target-relevant objects for ScanQA and ScanRefer is depicted in Fig 11. We instruct LLama3.1 based on the objects' spatial information and annotations from ScanQA and ScanRefer.

Generate reasoning-based segmentation data from object annotations in a 3D scene. Each object annotation is structured as follows: {instance_id: <id>, category: <category>, center: [<x>, <y>, <z>], sizes: [<length>, <width>, <height>], captions: [<captions>]}. Key points: 1. All measurements (coordinates and sizes) are in meters. 2. Objects with the same <ins_id> are considered identical.

Task requirements:
1. Generate challenging object-centric questions using the annotations:
    a. Focus on detecting and distinguishing objects. The answer should specify one or more objects.
    b. Include at least two of the objects' uses, appearance, positions, and sizes.
    c. Require logical reasoning to answer.
    d. Ensure questions are specific with one clear answer.
    e. Ensure object descriptions distinctly identify them from similar items.
2. Provide step-by-step reasoning for each question follow these steps:
    1. Analyze the properties and spatial relationship of objects in the scene.
    2. Analyze the question and identify question-related objects.
    3. Construct a logical step-by-step reasoning based on your analyzation to answer the question.
    4. Present the reasoning steps clearly.
    5. Use [<ins_id>] to refer objects in reasoning steps. For multiple objects of the same category, use [<ins_id>][<ins_id>][<ins_id>], etc.
3. Answer the question accurately.
4. Only use objects from the annotations.
5. Output format: 'question;reasoning;answer'. Don't output anything else.

Now generate for following annotations:

Figure 10: The prompt for yielding 3D ReasonSeg.

**Rule-based filtering.** In the context of rule-based filtering, we conducted a manual examination of the generated data and formulated filtering strategies. Firstly, we removed data containing phrases such as "in the right of" to mitigate potential ambiguity in 3D space. Secondly, for data involving spatial relationships like "near", "on", "under", etc., we calculated the distances between the center points of object point clouds and filtered out data with distances surpassing a predefined threshold. Thirdly, in instances of superlative relationships like "largest" or "nearest", we evaluated object center distances or sizes to validate their adherence to the criteria. Additionally, we excluded data that strayed from the specified prompt structure and identified and eliminated objects created by LLama3.1 based on point cloud instance labels. Furthermore, we pruned data exceeding a designated threshold of related objects and eliminated instances where each reasoning step yielded the same related objects to avoid redundancy. Lastly, we discarded data where reasoning conditions overlapped with reasoning results during inference and where the final step of reasoning conflicted with the answer to the question.

## B   ADDITIONAL QUALITATIVE RESULTS.

Additional visual qualitative results of 3D ReasonSeg are presented in Fig. 12. Each row illustrates the user instruction, prediction, target-relevant objects, and ground truth, respectively.

## C   TASK FORMULATION DETAILS

We introduce each instruction with the **USER:** designation and format all tasks as auto-regressive generation sequences under the **ASSISTANT:** label.

Generate step-by-step reasoning for question-answer pairs based on 3D scene object annotations. Each question-answer pair follows this structure: {question_id: <question_id>, question: <question>, answer: <answer>}. Object annotations are structured as: {instance_id: <instance_id>, category: <category>, center: [<x>, <y>, <z>], sizes: [<length>, <width>, <height>], captions: [<captions>]} Key points: 1. All measurements (coordinates and sizes) are in meters. 2. Objects with the same <ins_id> are considered identical.

Task requirements:
1. Process each question-answer pair, follow these steps:
    a. Analyze the properties and spatial relationship of objects in the scene.
    b. Analyze the question and identify question-related objects.
    c. Construct a logical step-by-step reasoning based on your analyzation to answer the question.
    d. Present the reasoning steps clearly.
    e. Use [<instance_id>] to refer objects in reasoning steps.
    f. Avoid mention the annotations.
2. Output format: '<question_id>;reasoning'. Don't output anything else.

Now generate for following annoatations:

---

List description-related objects in description of 3D scene object annotations. Object annotations are structured as: {annotation_id: <annotation_id>, instance_id: <instance_id>, category: <category>, description: <description>} Key points: 1. All measurements (coordinates and sizes) are in meters. 2. Objects with the same <ins_id> are considered identical.

Task requirements:
1. Analyze the properties and spatial relationship of objects in the scene based on the annotations.
2. Process each annotation, follow these steps:
    a. Analyze the description and identify description-related objects.
    b. Use [<instance_id>] to refer description-related objects.
3. Output format: '<annotation_id>;<description-related_objects>'. Don't output any other things.

Now generate for following annoatations:

Figure 11: The prompt for generating target-relevant objects for ScanQA and ScanRefer.

**3D Question Answering** tasks the model with responding to questions based on comprehensive knowledge of a 3D scene. We utilize a structured format such as: "**USER:** Given the 3D scene, please answer the question: [QUESTION]. **ASSISTANT:** [ANSWER]", where [QUESTION] and [ANSWER] are sourced from the ScanQA dataset.

**Scene Description** requires the model to describe the entire scene using natural language. We adopt a format like: "**USER:** Describe this scene. **ASSISTANT:** [DESCRIPTION]", where [DESCRIPTION] is derived from 3D-LLM.

**3D Instance Segmentation** task involves segmenting object categories within a scene. Our template is: "**USER:** Please segment the category in this scene. **ASSISTANT:** Sure, the segment results are [SEG]...[SEG]", where the [CATEGORY] is from ScanNet200 and each [SEG] corresponds to an individual object.

**3D Referring Segmentation** requires the model to segment a target object based on a brief description. Our template is: "**USER:** Please segment this object: [DESCRIPTION]. **ASSISTANT:** Sure, the segment result is [SEG]", where the [DESCRIPTION] is from ScanRefer and the [SEG] represents the segmentation mask of the target object.

**3D Reasoning Segmentation** necessitates understanding the visual attributes, functions, and relationships of objects in the query, followed by using reasoning to segment the target object. Our template is: "**USER:** Given the 3D scene, please answer the question with segmentation mask: [QUESTION]. **ASSISTANT:** [ANSWER]", where [QUESTION] and [ANSWER] are sourced from the 3D ReasonSeg dataset.

| Instruction | Prediction | Relevant Objects | Ground Truth |
|---|---|---|---|
| **USER:** What is the object that is used for sitting and located in the living room, near the tv and the coffee table? | | | |
| **USER:** What is the largest piece of furniture designed for sitting, next to the desk? | | | |
| **USER:** What is the object that is used for writing notes and is located near the window? | | | |
| **USER:** What is the object that is used for disposing of waste and is located near the counter? | | | |
| **USER:** What is the largest piece of furniture designed for sitting, next to the coffee table? | | | |
| **USER:** What is the largest piece of furniture designed for sitting, next to the coffee table? | | | |

16

Figure 12: More qualitative results.

**Spatial Relation Reasoning:**

**Step 1:**

**Question Answering Task and Reasoning Segmentation Task:**
- **USER:** Given the 3D scene, [QUESTION]. Please segment the question-related objects. **ASSISTANT:** [ANSWER]
- **USER:** [QUESTION] Please segment the question-relevant objects. **ASSISTANT:** [ANSWER]
- **USER:** Please segment the relevant objects for the question: [QUESTION]. **ASSISTANT:** [ANSWER]

**Referring Task:**
- **USER:** Please segment the relevant objects for this description: [DESCRIPTION]. **ASSISTANT:** [ANSWER]
- **USER:** This is a description of a object: [DESCRIPTION]. Please segment the description-related objects. **ASSISTANT:** [ANSWER]
- **USER:** Which objects related to this description: [DESCRIPTION]? Please segment it. **ASSISTANT:** [ANSWER]

---

**Step 2:**

**Question Answering Task:**
- **USER:** Given the 3D scene, [QUESTION]. Please answer the question. The question-related objects are [SEG]...[SEG]. You may need to pay attention to them. **ASSISTANT:** [ANSWER]
- **USER:** Answer this question: [QUESTION]. The question-relevant objects are [SEG]...[SEG]. You should pay attention to them. **ASSISTANT:** [ANSWER]

**Reasoning Segmentation Task:**
- **USER:** Given the 3D scene, [QUESTION]. Please answer the question with segmentation mask. The question-related objects are [SEG]...[SEG]. You may need to pay attention to them. **ASSISTANT:** [ANSWER]
- **USER:** Answer this question and output segmentation mask: [QUESTION]. The question-relevant objects are [SEG]...[SEG]. You should pay attention to them. **ASSISTANT:** [ANSWER]

**Referring Task:**
- **USER:** This is a description of a object: [DESCRIPTION]. Please segment this object. The description-related objects are [SEG]...[SEG]. You may need to pay attention to them. **ASSISTANT:** [ANSWER]
- **USER:** Please segment the object match to this description: [DESCRIPTION]. The description-relevant objects are [SEG]...[SEG]. You should pay attention to them. **ASSISTANT:** [ANSWER]

Figure 13: More template for Spatial Relation Reasoning.

**Spatial Relation Reasoning Templates.** The specifics of Spatial Relation Reasoning are delineated in Sec. 3.2, additional templates for SR$^2$ are depicted in Fig. 13, encompassing tasks such as question answering, reasoning segmentation, and referring.

