# OpenReview forum: "SR$^2$: BOOSTING 3D LARGE LANGUAGE MODEL WITH SPATIAL RELATION REASONING"
_ICLR.cc/2025/Conference — ICLR 2025 Conference Withdrawn Submission_

### Official Review · Reviewer_QxYa · 2024-10-21

**Soundness:** 3
**Presentation:** 3
**Contribution:** 3
**Rating:** 6
**Confidence:** 4

**Summary:**

This paper proposes a 3D reasoning segmentation method and a corresponding benchmark. It first proposes a baseline reasoning segmentation model following LISA. Then the base model is improved by the presented SRR to segment the target from coarse to fine. The authors collected data to train the model to first segment relevant objects and then segment the target by focusing on the priors. Experimental results on 3 benchmarks validate the effectiveness of the proposed method.

**Strengths:**

1. This paper is well-written and easy to follow.
2. The SRR framework is well motivated and interesting.
3. The performance of the method on three mainstream benchmarks is good.

**Weaknesses:**

1. The improvement after incorporating SRR is not so significant on most metrics according to Table 1. Considering this point, I think the efficiency of SRR should be provided, e.g., additional inference time, memory footprint, which can demonstrate a comprehensive tradeoff.
2. In Table1, there is no other method reported on 3D ReasonSeg benchmark. The authors should implement some representative methods on this for fair comparison.

**Questions:**

In line 169: "Subsequently, the Q-Former compresses the scene’s information into several latent queries $q_l$". What is the definition of $q_l$? Is it learnable parameters or extracted from the 3D representation?

---

### Official Review · Reviewer_GbgR · 2024-10-22

**Soundness:** 2
**Presentation:** 2
**Contribution:** 2
**Rating:** 5
**Confidence:** 4

**Summary:**

The paper proposes a new spatial relation reasoning method tailored for 3D scene understanding tasks and introduces the 3D ReasonSeg dataset. The spatial relation reasoning approach demonstrates potential effectiveness in enhancing scene understanding.

**Strengths:**

The spatial relation reasoning module employs a 2-step design tailored for 3D scene understanding, effectively capturing complex object relationships. The paper is commendably clear and easy to follow, with experiments validating the effectiveness of the proposed method.

**Weaknesses:**

The experimental results indicate that the improvement brought by $SR^2$ is relatively marginal. Specifically, the performance gain is only 0.1 on ScanRefer Acc@50 and 1.5 on the 3D ReasonSeg dataset.

Minor issue:
Inconsistent terminology: The $SR^2$ method is inconsistently referred to as SPR in L227 and L295.

**Questions:**

1. The viewpoint can heavily influence 3D object relationships, as the definition of 'left' and 'right' depends on the user's perspective. How do the $SR^2$ method and the 3D ReasonSeg dataset account for such viewpoint dependence? This is a core consideration in 3D scene understanding, especially regarding object relationships.
2. How do other 3D multi-modal large language models perform on the 3D ReasonSeg dataset?
3. Given that the pre-train dataset includes general datasets like ScanQA, ScanRefer, ScanNet200, 3D-LLM, and 3D ReasonSeg, how can we be sure that the performance superiority over other methods is not simply due to the varied pre-train datasets?
4. Can you provide some failure cases from the $SR^2$ method? These would help us better understand the characteristics and limitations of the $SR^2$ method.

---

### Official Review · Reviewer_3aZS · 2024-11-02

**Soundness:** 2
**Presentation:** 2
**Contribution:** 2
**Rating:** 5
**Confidence:** 3

**Summary:**

The paper proposes a new method to improve the spatial reasoning capability of 3D MLLM. The pipeline consists of two steps: identify all relevant elements first and then determine target among them. The authors have also set up a new benchmark named 3D ReasonSeg. They claim the proposed dataset can more comprehensively evaluate different models' capability in terms of complex spatial reasoning. Experiment have shown the proposed method improves the performance of base model on several datasets.

**Strengths:**

1. The pipeline makes sense for me. Intuitively, it would be good to decompose a complex spatial reasoning problem into 2 different stages, involving both coarse-grained and fine-grained steps.

2. The teaser figure is clear to demonstrate the paper's motivation and major method.

**Weaknesses:**

1. The authors have set up a new benchmark and claim that the proposed new benchmark can provide a more comprehensive evaluation in terms of the 3D spatial reasoning capability of the models. It would be better if the authors can have a table to summarise the different between the proposed dataset compared previous ones to make the contributions and differences more clear.

2. As in table 1, the improvement of adding SR^2 is not significant - only about 1% for most of the metrics. It would be more convincing if more improvement is brought by the proposed pipeline.

**Questions:**

I suggest the authors to address the questions raised in the weakness section during the discussion period

---

### Official Review · Reviewer_b441 · 2024-11-03

**Soundness:** 2
**Presentation:** 3
**Contribution:** 2
**Rating:** 5
**Confidence:** 4

**Summary:**

In this paper, the authors aim to strengthen relational reasoning capabilities in 3D environments. The Spatial Reasoning framework is proposed to mimic human reasoning behavior. A new benchmark is constructed for more specific training and evaluation.

**Strengths:**

- The problem studied in this paper, i.e., improving the 3D-LLM with spatial reasoning, is important and well-motivated.
﻿
- This paper is well-organized and easy to follow.
﻿
- Contributing a benchmark named 3D ReasonSeg for evaluation.

**Weaknesses:**

- In my view, the proposed framework is a two-stage sequential reasoning process, where stage one detects relevant objects and stage two reasons on these sampled objects. Such a pipeline is quite straightforward, lacking some technical contributions.
﻿
- I believe 3D spatial information such as 3D coordinate and geometry information is fundamental in distinguishing 3D tasks from 2D tasks. However, how to better leverage such 3D spatial information to improve 3D-LLM's spatial reasoning is still unexplored.
﻿
- Fig.1 is somewhat blurry, making it difficult to distinguish the objects clearly.

- Besides the positional relationships between objects, I believe the geometric shapes and relative sizes of objects at varying scene scales are also crucial for 3D spatial reasoning, which is ignored in this work.

**Questions:**

- The supplementary materials should be in a separate file, but the author seems to have included them at the end of the main file.

---

### Official Review · Reviewer_J4fm · 2024-11-04

**Soundness:** 3
**Presentation:** 3
**Contribution:** 2
**Rating:** 5
**Confidence:** 3

**Summary:**

The authors proposed a spatial relation reasoning method to tackle the problem of point-cloud reasoning task. Instead of doing reasoning in a one-stage end2end manner, the authors adopt a strategy of first get the target-relevant objects in point-cloud and then reason the relationships between the target objects. The experiment results demonstrate the effectiveness of the proposed method.

**Strengths:**

The motivation is good. Indeed directly reasoning everything is hard due to the lack of dataset and we should decompose the complex reasoning tasks into simpler tasks. The paper is also well-written. The experiment result also demonstrates that the effectiveness of the results.

**Weaknesses:**

Based on above strength especially about the motivation, I would say however the proposed method seems to be too heavy. I like motivation that first locate the objects then infer the relationship. But I think the method looks very heavy and redundant. it seems that it does not necessarily call the heavy 3D-VLLM twice. It should be able to directly run an efficient 3D VLLM to locate the objects then leverage the localized 3D position for directly reasoning the relationship instead of using complex tokens from features.

Besides, if just look at baseline vs. baseline + SR2, the proposed method does not improve the performance significantly. I would also attribute the slight improvement to the redundant design since maybe the super-point grouping introduce more noisy information. More importantly, I found that the baseline the authors use already achieves very significant improvement compared to other methods. In that case, it seems that using better LLM and more advanced vision encoders are more important compared to the motivation of decomposition.

I would also recommend the author compared the latency for all the experimented baselines. Again, I like the motivation so I do expect that with the new proposed "two-phase paradigm", we can use more efficient models to achieve better performance instead of simply calling a heavy model twice while not improving much performance.

**Questions:**

See the weakness. I especially expect the authors can address my concerns about the motivation and the trade-off between the efficiency and the performance.

---

### Note · Authors · 2024-11-13

I have read and agree with the venue's withdrawal policy on behalf of myself and my co-authors.